



# XBT data collected along the Southern Ocean "chokepoint" between New Zealand and Antarctica, 1994-2024

Giuseppe Aulicino*[,1,2], Antonino Ian Ferola[1], Laura Fortunato[1], Giorgio Budillon[1,3], Pasquale Castagno[3,4], Pierpaolo Falco[3,5], Giannetta Fusco[1,3], Naomi Krauzig[3,5], Giancarlo Spezie[1,3], Enrico Zambianchi[1,3], Yuri Cotroneo*[,1,3]

[1] Dipartimento di Scienze e Tecnologie, Università degli Studi di Napoli "Parthenope", Napoli, 80143, Italy
[2] Istituto di Scienze Polari, Consiglio Nazionale delle Ricerche, Bologna, 40129, Italy
[3] Consorzio Nazionale Interuniversitario per le Scienze del Mare (CoNISMa), Roma, 00196, Italy
[4] Dipartimento di Scienze Matematiche e Informatiche, Scienze Fisiche e Scienze della Terra, Università degli Studi di Messina, 98122, Italy
[5] Dipartimento di Scienze della Vita e dell'Ambiente, Università Politecnica delle Marche, Ancona, 60131, Italy

* *Correspondence to:* Giuseppe Aulicino (giuseppe.aulicino@uniparthenope.it); Yuri Cotroneo (yuri.cotroneo@uniparthenope.it)

**Abstract.** This study presents the water column temperature data collected during several cruises on board the Italica, Araon and Laura Bassi research vessels, in the framework of the Climatic Long-term Interaction for the Mass balance in Antarctica (CLIMA), Southern Ocean Chokepoints Italian Contribution (SOChIC), and Marine Observatory of the Ross Sea (MORSea) projects, funded by the Italian National Antarctic Research Program (PNRA). Data were collected between New Zealand and the Ross Sea during the austral summers from 1994/1995 to 2023/2024. Across this chokepoint of the Antarctic Circumpolar Current, XBT Sippican T7 probes were launched with a regular 20 km sampling, providing temperature profiles with a vertical resolution of 65 cm and a maximum nominal depth of 760 m. All temperature profiles underwent a rigorous quality control, including a general malfunctioning verification, the removal of spikes, the consistency check of adjacent profiles, the comparison to regional oceanographic features and satellite altimetry observations, and a final visual check by operator. Data quality checks led us to discard about 12% of acquired XBT measurements. This dataset contributes to the improvement of our understanding of Southern Ocean features, being highly valuable for studies focusing on climate variability, especially across the Antarctic Circumpolar Current and its fronts. Furthermore, we expect that the collected XBT data will serve as a useful tool for the calibration and validation of recent satellite observations and for the improvement of Southern Ocean oceanographic simulations.



## 1 Introduction

The temperature of the ocean is one of the key parameters identified by the Global Climate Observing System (GCOS) as being essential for climate studies (World Meteorological Organization, 2016). Together with salinity values, ocean temperatures are necessary to identify and trace the main water masses and monitor their evolution at different spatial and temporal scales.

On the larger scales, collecting oceanic temperature and salinity data is of paramount importance to the study of the global thermohaline circulation, which plays a pivotal role in Earth's climate system. The Southern Ocean (SO) plays a fundamental role in this circulation (Gille, 1994; Rintoul, 2018), as some of the global thermohaline circulation "engines" are located near the Antarctic coast, associated with polynya areas (Morales Maqueda et al., 2004; Aulicino & Wadhams, 2022). At smaller scales, temperature data can be used to describe the vertical structure of the ocean (e.g., the thermocline depth and its variability), to locate fronts between different water masses, determine the ocean heat content and volume transport, and to identify meso- and sub-mesoscale ocean dynamics. The main current in the SO is the Antarctic Circumpolar Current (ACC), which is its primary source of heat, nutrients and momentum (Sokolov & Rintoul, 2009a, 2009b). The ACC is one of the largest currents on the planet, flowing from west to east and isolating the Antarctic continent, which makes it strongly dependent on the SO conditions. Additionally, the Antarctic ecosystem is very fragile and temperature-dependent, which highlights the importance of monitoring physical changes in the ocean that surrounds it (Convey & Peck, 2019). Therefore, monitoring the SO and its temperature is essential for improving our knowledge of the processes driving the Antarctic variability and the global climate balance (Rintoul, 2018; Armour et al., 2016).

Despite its importance, SO has consistently faced a scarcity of in situ observations due to its remote location and the extreme weather conditions, which often hinder research activities to be carried out on site. The measurements are further limited by the seasonal sea ice presence that inhibits the navigation and the data collection. Additionally, in situ data collection is often conducted with instruments and probes used from ships travelling at their normal speed (e.g., Expendable BathyThermographs – XBT), without the possibility to perform classical full depth CTD casts that require ship stops. The advent of the international ARGO program increased significantly the number of hydrographic observations available in the SO throughout all seasons (Roemmich et al., 2022). However, Lagrangian floats do not allow the collection of information along repeated monitoring lines.

Accordingly, many steps have been taken over time to obtain ocean temperature data through remote sensing. Satellite data provide valuable insights about the upper ocean, especially when considering that the surface layer is closely related to fundamental phenomena (e.g., ocean-atmosphere physical



and biogeochemical interactions, fronts, currents, meanders, eddies) impacting the large-scale
circulation and the meso- and small-scale characteristics of the ocean (e.g., McGillicuddy, 2016;
Cotroneo et al., 2016; Seo et al., 2023). Additional information about the water column can also be
retrieved from numerical models (e.g., Downes et al., 2015) and 3D reconstructions inferred through
machine learning and statistical techniques applied to satellite observations, such as sea surface
temperature (e.g., Buongiorno Nardelli et al., 2020). Nonetheless, in-situ measurements are
indispensable for achieving the necessary precision and depth coverage. In addition, they provide
critical ground-truth for the calibration and validation of satellite retrievals of surface variables, and
the improvement of data acquisition algorithms (Aulicino et al., 2022). It is therefore evident that the
collection of in-situ data is essential for monitoring ocean temperature.
To this aim, the University of Naples Parthenope has been taking part since 1994 in the organization
and execution of several oceanographic campaigns along the PX36 monitoring line in the Pacific
sector of the SO, i.e., between New Zealand and the Ross Sea, in the framework of the Italian National
Antarctic Research Program (PNRA). During each expedition, XBT launches were carried out,
collecting ocean temperature data from surface to a maximum of about 760m depth (Falco et al.,
2022). This study presents the collected XBT dataset, which significantly contributes to the
accessibility of extensive ocean temperature data.
In this paper, the methodologies used for data collection and quality control (QC) are described in
Section 2; the results and the discussion are reported in Section 3; the data record details and the
conclusions are summarized in Section 4.



## 2 Data and methods

### 2.1 The XBT dataset

An XBT system is composed of several key components: an expendable ballistic probe that descends into seawater; a data acquisition device that records an electrical signal and converts it into usable numerical data (with the support of a computer unit); a double copper wire that connects the falling probe to the acquisition device (Goni et al., 2019; Parks et al., 2022; Simoncelli et al., 2024). As the probe descends through the water column, temperature measurements are acquired using a Negative Temperature Coefficient (NTC) thermistor mounted on the probe zinc nose, which alters its resistance in response to the seawater temperature it comes into contact with. The insulated copper wire is unwound simultaneously by two spools, i.e., clockwise on the ship and counterclockwise in the falling probe. This technique decouples the XBT vertical descent through the seawater from the ship translational motion (Simoncelli et al., 2024). Data recording continues until the wire breaks or the recording is terminated by the operator. The depth associated with a temperature measurement is not sensed directly because XBT probes do not contain pressure sensors. Instead, it is estimated using a phenomenological Fall Rate Equation (FRE) provided by the manufacturer, with coefficients that vary based on the probe type.

The uncertainties on temperature and pressure values make the XBT probe accuracy be generally rated to ± 0.10°C (Parks et al., 2022), although differences can be retrieved depending on the manufacturer and the manufacturing date of different devices (Cowley and Krummel, 2022). Consequently, some crucial information should be always provided with any XBT dataset for subsequent optimal use of the measurements, including a complete description of the system characteristics in the metadata (e.g., probe type, fall rate coefficients, data originator, platform).

We present here the dataset of water column temperatures collected in the Pacific sector of the Southern Ocean through XBT casts during several research cruises on board the Italian research vessels "Italica" and "Laura Bassi" and the Korean icebreaker "Araon" (see Table 1). These activities were carried out in the framework of the Italian PNRA by several scientific projects, e.g., Climatic Long-term Interaction for the Mass balance in Antarctica (CLIMA), Southern Ocean observing system and Chokepoints Italian Contribution (SOChIC) and Marine Observatory in the Ross Sea (MORSea).

The XBT casts were carried out during the austral summers between 1994/1995 and 2023/2024, mainly in January and February (Figure 1), using Sippican T7 probes providing temperature profiles with a vertical resolution of 65 cm and a maximum nominal depth of 760 m. Only during the 1994/1995 (PNRA X) and 1995/1996 (PNRA XI) cruises some Sippican T5 probes were used,



reaching a maximum depth of 1830 m, as reported in the campaign metadata information (Table 2).
The majority of transects were completed in 5-6 days and provide a synoptic picture of the thermal
structure of the upper SO across its Pacific Sector (Figure 2). A regular 20 km sampling rate was
adopted with occasional increased sampling frequency over the main frontal regions of the ACC.

**Table 1.** List of scientific cruises included in this dataset carried out between November 1994 and January 2024

| Cruise name | R/V | Start date | End date | Latitude | Longitude |
|---|---|---|---|---|---|
| PNRA X | ITALICA | 03 November 1994 | 02 March 1995 | 47.00 - 74.99°S | 172.02°E - 175.90°W |
| PNRA XI | ITALICA | 07 January 1996 | 18 February 1996 | 48.66 - 72.01°S | 173.56°E - 179.79°E |
| PNRA XII | ITALICA | 26 January 1997 | 19 February 1997 | 46.17 - 74.69°S | 166.24°E - 179.82°E |
| PNRA XIII | ITALICA | 23 November 1997 | 06 March 1998 | 46.25 - 72.71°S | 171.39°E - 179.43°W |
| PNRA XIV | ITALICA | 05 January 1999 | 11 January 1999 | 48.07 - 69.00°S | 173.70°E -178.55°E |
| PNRA XV | ITALICA | 07 January 2000 | 18 February 2000 | 49.17 - 69.83°S | 173.13°E - 178.41°E |
| PNRA XVI | ITALICA | 06 January 2001 | 26 February 2001 | 48.75 - 75.94°S | 170.59°E - 179.72°E |
| PNRA XVII | ITALICA | 24 December 2001 | 31 December 2001 | 48.50 - 69.30°S | 160.39°E - 178.01°E |
| PNRA XVIII | ITALICA | 06 January 2003 | 11 January 2003 | 48.00 - 71.26°S | 172.93°E - 177.47°E |
| PNRA XIX | ITALICA | 24 December 2003 | 28 December 2003 | 46.36 - 66.17°S | 173.81°E - 179.99°E |
| PNRA XX | ITALICA | 01 January 2005 | 06 January 2005 | 48.03 - 70.49°S | 174.22°E - 178.38°E |
| PNRA XXI | ITALICA | 01 January 2006 | 04 January 2006 | 48.03 - 66.50°S | 174.59°E - 179.93°E |
| PNRA XXII | ITALICA | 05 February 2007 | 10 February 2007 | 47.23 - 71.99°S | 170.86°E - 174.26°E |
| PNRA XXIII | ITALICA | 16 January 2008 | 21 January 2008 | 47.50 - 68.99°S | 174.18°E - 178.63°E |
| PNRA XXV | ITALICA | 25 January 2010 | 29 January 2010 | 46.38 - 70.00°S | 173.63°E - 178.00°E |
| PNRA XXVII | ITALICA | 13 January 2012 | 19 January 2012 | 47.85 - 65.96°S | 172.03°E - 176.54°E |
| PNRA XXVIII | ARAON | 24 January 2013 | 06 February 2013 | 47.20 - 68.5°S | 158.30°E - 177.00°E |
| PNRA XXIX | ITALICA | 30 December 2013 | 18 February 2014 | 48.01 - 78.83°S | 167.07°E - 175.84°W |
| PNRA XXX | ARAON | 02 January 2015 | 10 January 2015 | 47.99 - 73.22°S | 157.02°E - 173.81°E |
| PNRA XXXI | ITALICA | 16 January 2016 | 28 January 2016 | 47.49 - 72.40°S | 171.56°E - 175.00°E |
| PNRA XXXII | ITALICA | 31 December 2016 | 05 January 2017 | 48.01 - 68.77°S | 174.09°E - 179.85°W |
| PNRA XXXIV | ARAON | 08 February 2019 | 12 February 2019 | 47.99 - 69.75°S | 166.79°E - 170.87°E |
| PNRA XXXV | LAURA BASSI | 07 January 2020 | 12 January 2020 | 48.01 - 69.25°S | 172.97°E - 178.84°E |
| PNRA XXXVI | LAURA BASSI | 25 December 2020 | 02 January 2021 | 46.96 - 73.39°S | 172.82°E - 175.89°E |
| PNRA XXXVII | LAURA BASSI | 08 January 2022 | 26 January 2022 | 47.54 - 76.35°S | 171.20°E - 177.58°W |
| PNRA XXXVIII | LAURA BASSI | 06 January 2023 | 12 January 2023 | 46.56 - 72.27°S | 169.40°E - 178.70°E |
| PNRA XXXIX | LAURA BASSI | 07 January 2024 | 12 January 2024 | 48.20 - 70.00 °S | 166.30 °E – 176.40°E |




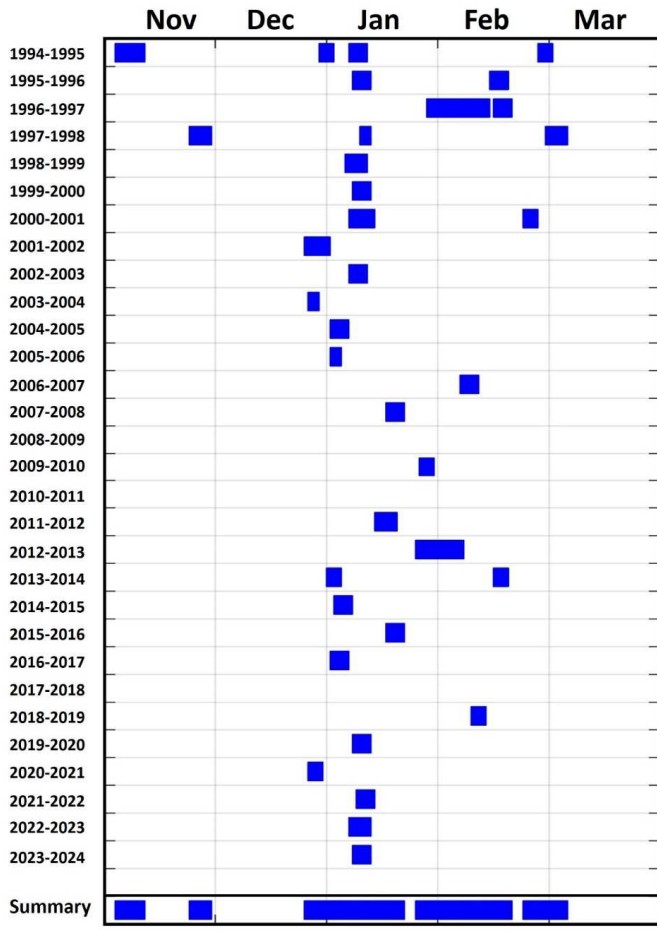

**Figure 1.** Temporal distribution of the oceanographic campaigns conducted along the New Zealand-Antarctica "chokepoint" between 1994 and 2024.

**Table 2.** Characteristics of the different XBT probes used in this study: nominal depth guaranteed by Sippican; maximum ship speed suggested by Sippican for an optimal drop; coefficients of Fall Rate Equation $D(t) = At - Bt^2$ used for depth calculation provided by the manufacturer; amount of ZAMAK, copper and plastic for each probe type (adapted from Simoncelli et al., 2024)

| Probe type | Max rated depth (m) | Max ship speed (knots) | FRE coeff A (ms-1) | FRE coeff B (ms-1) | ZAMAK (kg) | Plastic (kg) | Copper (kg) |
|---|---|---|---|---|---|---|---|
| **Sippican T5** | 1830 | 6 | 6.828 | 0.00182 | 0.613 | 0.125 | 0.357 |
| **Sippican T7** | 760 | 15 | 6.691 | 0.00225 | 0.576 | 0.052 | 0.240 |


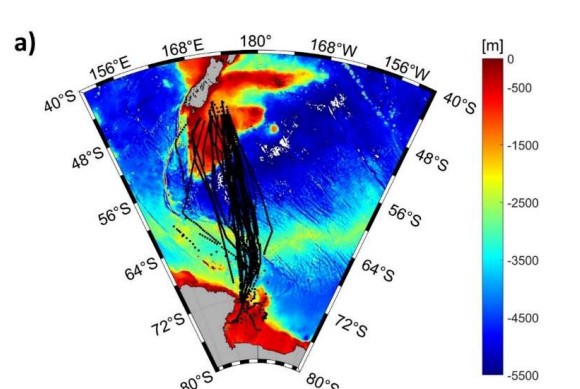 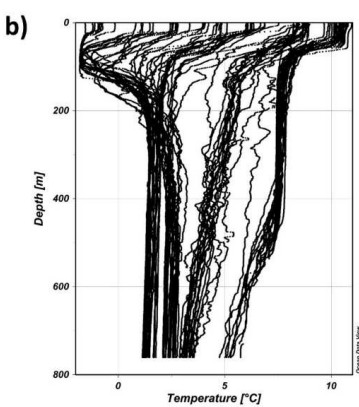


**Figure 2. a)** Map of the Southern Ocean area between New Zealand and Antarctica. The black dots represent the position of all XBT launches carried out between December 1994 and January 2024. **b)** An example of temperature vertical profiles collected through XBT across the New Zealand – Ross Sea chokepoint during the XXXV Italian Antarctic Expedition.



### 2.3 Quality Control

Various types of malfunctions can affect XBT measurements and result in inaccurate temperature readings within the temperature profile. These faults can appear as a spike in a single recorded value or affect the temperature across a range of depths. Moreover, some issues can create errors that mimic real phenomena, such as temperature inversions or fronts (Parks et al., 2022). Sometimes, profiles can be corrected by deleting or filtering sections of the original data. However, an accurate quality control procedure must be implemented before any data is discarded or manipulated. Additionally, a flagging scheme is generally applied to provide XBT dataset users with quality indicators of the oceanographic data.

Quality flags (QFs) are essential for enabling users to filter the XBT dataset according to the specific quality requirements for the intended use. Several flagging scheme exist in agreement with recommendations provided by the Intergovernmental Oceanographic Commission of UNESCO (IOC, 2013). In this study we follow the suggestions provided by the Global Temperature and Salinity Profile Program (GTSPP) of the NOAA-NCEI (https://www.ncei.noaa.gov/products/global-temperature-and-salinity-profile-programme) resulting in the flagging scheme summarized in Table 3 for indicating the quality of each temperature and depth data point.






**Table 3.** The Quality Flags (QF) assigned to the XBT data

| QF | Quality | Description |
|----|---------|-------------|
| 0 | No QC | No quality control has been performed on this data. |
| 1 | Good data | The data is good. No malfunctions have been identified and consistency with real phenomena has been verified. |
| 2 | Probably good data | Minor malfunctions present which are small or correctable without affecting overall data quality. Some features (probably real) are present but these are unconfirmed. |
| 3 | Probably bad data | Data are suspect and present unusual features which are inconsistent with real phenomena, Data remains potentially correctable. |
| 4 | Bad data | The data appears erroneous. Evident errors are identified and there is no likelihood of correction. |



The assignment of QFs is the result of a series of quality control (QC) tests for both temperature and
depth data which are used to get a reliable quality check of the temperature measurements collected
through our XBTs and of the retrieved depths. Results of each test allowed to insert the relative flag
to the corresponding measurement according to the scheme shown in Table 3. QF=1 is assigned when
all the tests pass and QF=4 when at least one test fails. For temperature, more detailed checks are
performed, including a final visual check, allowing us to introduce QF=2 and QF=3 for probably
good and probably bad data, respectively (as detailed below).
Overall, the QC procedures applied to our dataset follow recommendations previously suggested by
NOAA, developed and refined in the last three decades (Bailey et al., 1994; Daneshzadeh et al.,1995;
Cowley and Krummel, 2022; Parks et al., 2022; Tan et al., 2023). These procedures include several
steps undertaken in a top-down manner, as temperature data are measured from the surface down,
and faults that occur at a given depth may impact on deeper data (Parks et al., 2022).
First, each XBT profile was tested for invalid metadata information, such as the correct time, cast
position and any other possible operator errors, using a sequence of independent checks. All identified
errors in date and time were corrected accordingly, with the support of the XBT launch clipboards
provided by operators on board. No errors were found concerning the position of the casts after the
comparison of latitudes and longitudes against gridded GEBCO 2 x 2 minutes bathymetry (GEBCO
Compilation Group, 2023). The check of unrealistic positions was also performed using the
calculation of vessel speed from profile date and time and an upper general threshold of 20 knots
(since most of the launches are realized by ships travelling in the range of 10/15 knots). Additionally,
the depth values of each XBT profile were compared to the last good depth value provided by the
operators (QF=1 is assigned to shallower depth values, otherwise they are flagged as QF=4).



Then, all the vertical temperature profiles were checked for nominal maximum depth (760 m), and
carefully inspected to identify malfunctions, coherence to regional oceanographic features, drop-to-
drop consistency along the cruise track, and presence of unusual features. In this context, the main
difficulty is usually found in distinguishing a common malfunction from a regional oceanographic
feature (i.e., unexpected increase of temperature southward or along the water column).
Consequently, unusual features were cross-validated by comparison to repeated (within 15 minutes)
or neighbouring profiles from the same voyage and eventually to available Austral summer ARGO
observations over the study area. To this aim, we took again advantage of XBT launch clipboards, in
which operators notified any instrument malfunctions, adverse weather conditions, sea ice presence
and local bottom depth. In particular, the bottom depth was relevant to constraining XBT data profiles
at the right elevations, especially when approaching shallow waters (QF=1 is assigned to values
shallower than bottom depth, otherwise they are flagged as QF=4). When the clipboard was not
available, we relied instead on the GEBCO 2 x 2 minutes bathymetry (GEBCO et al., 2023), which
resulted the most correspondent to the in situ reported depths over the area and period of study.
Additionally, a gross filter was applied to all the XBT profiles using a series of temperature thresholds
that vary on four vertical layers, as reported in Table 4. The thresholds were defined through the use
of ARGO data collected in the study area between 2004 and 2023. QF=4 was applied to data
exceeding the thresholds of ±0.5°C.

**Table 4.** Temperature thresholds applied to XBT profiles, defined in four levels.

| Depth range (m) | Temperature minimum (°C) | Temperature maximum (°C) |
|---|---|---|
| **0 - 100** | -1.866 | 14.698 |
| **100 - 250** | -1.865 | 11.093 |
| **250 - 500** | 0.068 | 8.717 |
| **500 - 760** | 0.826 | 8.266 |



Several studies assess that the XBT measurements near the sea surface may be considered unreliable
due to the stabilization of motion and thermal adaptation to the surrounding environment (e.g. Bailey
et al., 1994; Cowley and Krummel, 2022; Simoncelli et al., 2024). They also suggest that the first
acceptable value is at about 4 m depth and that the data user must be carefully informed in order to
exclude suspect surface values from scientific analyses. Here, we opted for providing all the original
measurements annotating their quality, as resulting from a dedicated test on the initial part of each
profile. This test calculates the differences between the value recorded at time t = 0.6 s (about 4 m



depth) and shallower measurements, classifying them based on the standard uncertainty on
temperature attributable to an XBT probe (0.10 °C) as a metric (Simoncelli et al., 2024). Therefore,
temperature data are assigned QF=1 if the difference is less than or equal to standard deviation (std);
QF=2 if it is comprised between std and 2*std; QF=3 if it is comprised between 2*std and 3*std; and
QF=4 if it is higher than 3*std.
Then, the XBT profiles were examined for the presence of spikes, unrealistic oscillations and
unusually gradients in temperature data, as well as sharp variations toward negative or higher values,
which could be caused by copper wire breaks. Data are mostly flagged as good (QF=1) or bad (QF=4)
values. Nonetheless, suspect data are compared with neighbouring profiles and ARGO climatology
over the study area, eventually assigning QF=1, QF=2 and QF=3 attributes. For example, QF=2 is
used when an XBT profile presents a step-like feature that is not confirmed by a neighbouring profile
but is consistent with similar features previously observed in the study region. QF=3 is used, instead,
when XBT values exhibit suspect temperature values that cannot be confirmed by a neighbouring
profile and occur in areas where there is no evidence of mesoscale structures (e.g., eddies or fronts).
Nevertheless, an increase or decrease in temperature over large depth ranges compared to
neighbouring profiles, can be also associated to an eddy, a frontal area or an intense current system.
Therefore, QF=1 is applied when repeated profiles showing similar temperatures or archive data can
confirm the feature. The larger scale description of ocean dynamics obtained through satellite
altimetry was also used for controversial results to identify the presence of eddies and frontal systems
affecting the temperature data.
However, some profiles might exhibit anomalous features that the described QC procedure could not
detect as erroneous values. Therefore, an additional visual check was carried out for each individual
cruise track and each vertical temperature profile to verify the assigned QF=2 and QF=3 flags and
identify any residual anomalies in the positioning of the XBT launches or outliers in the data
collection. This control was performed using the Ocean Data View (ODV) software (Schlitzer, 2023).
Overall, the entire QC led us to discard about 12% of acquired XBT observations, which were flagged
as bad or probably bad data (Figure 3).
**2.4 XBT data biases correction**
Previous studies assessed that temperature biases and depth errors, due to inaccurate time conversion
to depth through FRE, may affect XBT observations (e.g., Gouretski and Reseghetti, 2010; Cowley
et al., 2013). Although a full comprehension of the origins of these issues is still pending, several
experiments tried to quantify this bias by comparing XBT profiles with co-located CTD observations,
demonstrating that XBT temperatures are usually warmer than reality (Gouretski and Reseghetti



2010; Cheng et al., 2014). Different possible causes of biases emerged, including mechanical (e.g., probe type, manufacturer, year), external (e.g., launch height, meteo-marine conditions) and electrical (e.g., thermistor, wire) factors (Seaver and Kuleshov 1982; Green, 1984; Reverdin et al. 2009). Additionally, a decrease in fall rate was observed in cooler waters because of increased viscosity (Gouretski and Reseghetti 2010), making FRE corrections in the Southern Ocean extremely important (Cheng et al., 2014).

To address these problems, several correction schemes have been proposed over the past few decades. A comprehensive list of related papers is available at https://www.ncei.noaa.gov/products/xbt-corrections. Taking advantage of more than 220,000 XBT-CTD side-by-side pairs, Cheng et al. (2014) examined and compared existing methodologies, proposing a new correction scheme for historical XBT data for nine independent probe-type groups. Their study confirmed that depth error and pure temperature bias are temperature-dependent and may be influenced by the data acquisition and recording system. Moreover, the resulting scheme also considers that some biases affecting the XBT-derived temperature profiles vary with manufacturer/probe type and have been shown to be time dependent, and that depth correction varies with depth (Cheng et al. 2016).

In our dataset, we apply this methodology, which includes corrections for both temperature and depth values based on calendar year, water temperature, and probe type, to provide bias-corrected XBT measurements (Cheng et al., 2014). A full description of the methodology and an update table of applied coefficients are available at https://www.nodc.noaa.gov/OC5/XBT_BIAS/ch-method.html and http://www.ocean.iap.ac.cn/pages/dataService/dataService.html?navAnchor=dataService.

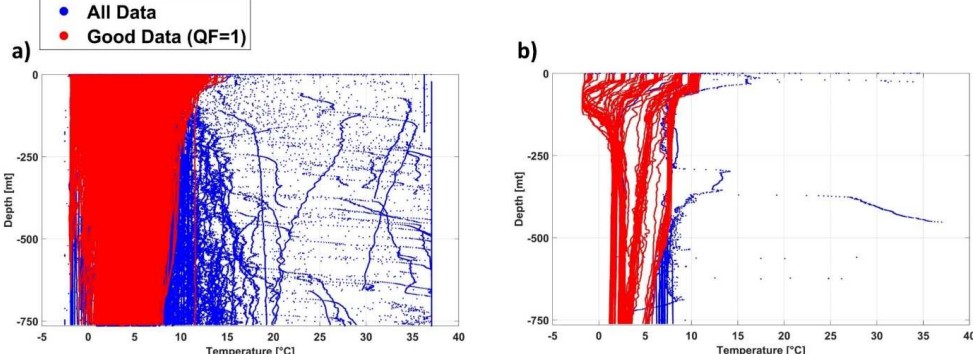

**Figure 3. a)** XBT observations collected between December 1994 and January 2024 over the New Zealand – Ross Sea chokepoint before (blue) and after (red) the quality check; **b)** An example of the quality check on the XBT data collected during the XXXV PNRA cruise.



## 3. Results and discussion

We believe this exceptional temperature dataset provides a valuable reservoir of high-resolution, independent, and trustworthy information. The dataset assumes notable significance, representing an extensive temporal series of data collected nearly every austral summer over the last 30 years, within the same oceanic sector of the SO and along the same monitoring transect (PX36). We exploited this information to provide 36 vertical sections of the ocean temperature, from the surface to about 800 m depth, along the New Zealand–Antarctica "chokepoint". Figures representing the latitudinal sections of corrected XBT temperatures during each leg are available in the supplementary information (Figures S1) .

The repeated temperature sections significantly enhance our understanding of ACC fronts and their evolution over the last three decades. A first application of the dataset is shown in Figure 4 where XBT observations collected during the XVIII PNRA expedition are used for the identification of the main ACC fronts positions: Northern Sub Antarctic Front (NSAF); Southern Sub Antarctic Front (SSAF); Polar Front (PF); Southern Antarctic Circumpolar Current (sACCf). The criteria used for identifying the fronts (Table 5) follow Budillon and Rintoul (2003), which compiles several hydrographic definitions (Botnikov, 1963; Belkin, 1990; Orsi et al., 1995; Rintoul et al., 1997). The SBdy of the ACC, usually described as the maximum southern extent of vertical maximum of T>1.5°C at about 200 m (Orsi et al., 1995), is not described in this sector as its position is coincident with the sACCf position in most of the available temperature sections.

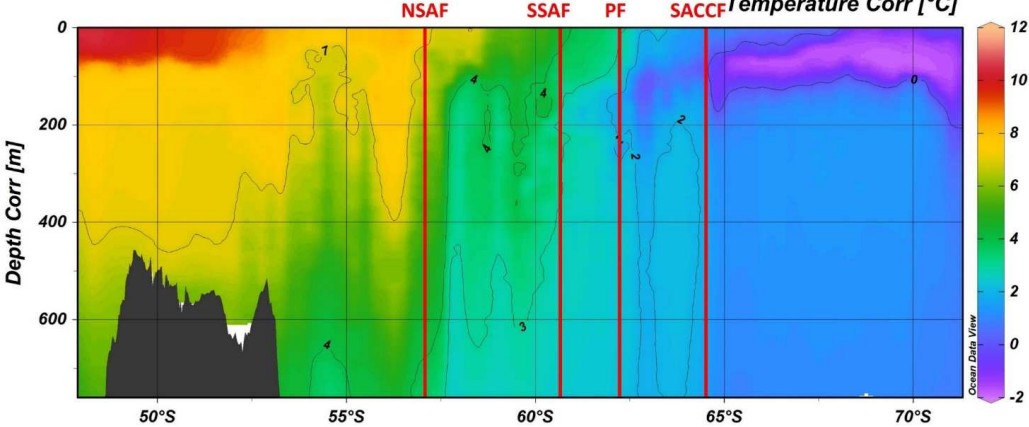

**Figure 4**. Temperature vertical section from XBT data collected during the XVIII PNRA expedition along the New Zealand–Antarctica "chokepoint" in which the vertical red lines represent the ACC main fronts positions: Northern Sub Antarctic Front (NSAF); Southern Sub Antarctic Front (SSAF); Polar Front (PF); Southern Antarctic Circumpolar Front (SACCF). The dark grey mask represents the bathymetry.




**Table 5.** Criteria for front definitions (Adapted from Budillon & Rintoul, 2003)

| Front | Definition | Reference |
|-------|-----------|-----------|
| Southern Antarctic Circumpolar Current Front (sACCf) | T > 1.8°C along the Tmax at depth > 500 m, farther north; T < 0°C along the Tmin at depth < 150 m, farther south. | Orsi *et al.* 1995. |
| Polar Front (PF) | T < 2°C at 200 m, farther south. | Botnikov 1963, Orsi *et al.* 1995. |
| Subantarctic Front (SAF) | Maximum temperature gradient in the range 3–8°C at 300 m. | Belkin 1990. |
| Northern Sub-Antarctic Front (NSAF) | Maximum temperature gradient in the range 4–7°C at 300 m. | Rintoul *et al.* 1997. |
| Southern Sub-Antarctic Front (SSAF) | Maximum temperature gradient in the range 3–4°C at 300 m. | Rintoul *et al.* 1997. |



The ACC fronts positions retrieved through XBT data also serve as ground truth for the validation of
those retrieved through satellite altimetry (e.g., Sokolov and Rintoul 2009a, 2009b; Graham et al.,
2012; Chapman, 2017), thereby enhancing the identification process of fronts within the SO. This is
extremely desirable in region characterized by significant influenced by topographic steering, such
as the area south of New Zealand where the presence of the Campbell Plateau severely affects the
ACC path (Figure 5). To point out differences and similarities between ACC fronts positions
identified through XBT and satellite observations, in Figure 5 we present a Sea Surface Height (SSH)
map of the study area, averaged over the period covered by the temperature section in Figure 4 (about
7 days). To identify the ACC fronts from satellite data, we applied the SSH isolines methodology that
associates a specific value of SSH with each front. For the selection of these values, we relied on
previous studies (Sokolov and Rintoul 2007, 2009a, 2009b) proving that the multiple jets of ACC
fronts are consistently aligned with streamlines identified by nearly constant circumpolar values of
SSH contours.
Furthermore, ACC fronts exhibit instabilities that give rise to the generation of eddies.
Eddies, characterized as vortices pervading the ocean, assume a pivotal role, particularly within the
SO, contributing significantly to the transfer of heat, nutrients, and momentum (e.g., Chelton et al.
2011a; Falco and Zambianchi, 2011; Cotroneo et al., 2013; Trani et al., 2014; Rintoul, 2018; Menna
et al., 2020). While altimetry proves valuable in gaining insights into surface eddy dynamics, it cannot
provide information regarding vertical temperature variations within the eddy structure. Through the
temperature sections derived from XBT data, we can discern the presence or absence of an eddy and
get basic observations for the analysis of its heat content.
An example is provided in Figure 6 where we present the latitudinal section of temperatures observed
during the return leg of the 2013-2014 Italian Antarctic expedition (PNRA XXIX). This section shows



the intrusion of a cold core eddy at about 53°S, next to the Campbell Plateau edge. The eddy is
characterized by a maximum negative temperature anomaly (eddy's core) of about -4°C with respect
to the surrounding water. This negative anomaly results in the formation of a depression in the SSH,
also detectable in satellite imagery. In the SSH map shown in Figure 7, the cold core eddy is identified
as a closed circle of the blue isoline associated with the SSAF.

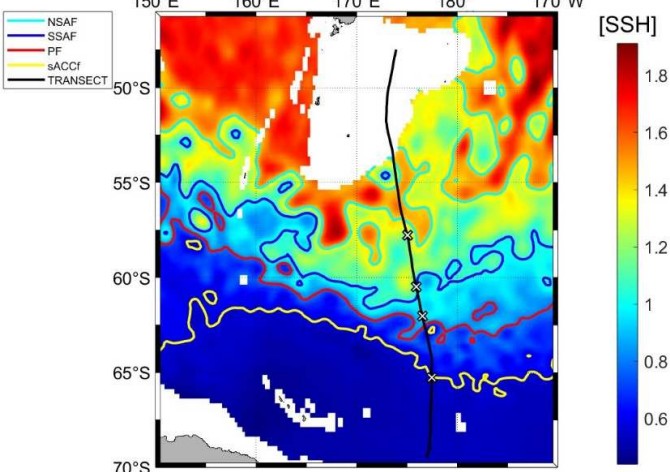


**Figure 5.** Altimetric map of SSH mediated throughout the XVIII PNRA expedition along the PX36 monitoring line.
Contours of different colours identify the position of the main fronts of the ACC retrieved through SSH: NSAF in cyan;
SSAF in blue; PF in red and sACCf in yellow. White crosses represent the position of the fronts derived from XBT data.
The ship's route is represented by the black line.

Generally, the combined use of in situ observations and satellite data is crucial as it prevents errors
in front positioning and eddy identification. Strong horizontal temperature gradients, often linked to
eddies, could be misinterpreted as ACC fronts. Similarly, this approach allows us to distinguish eddies
from other mesoscale structures, a difficult task when relying only on altimetry. XBT and satellite
information are also complementary in providing valid terms of comparison, at different temporal
and spatial scales (XBT at fine-scale; altimetry at meso- and large-scale), for numerical model
products representing ocean circulation and eddies dynamics (e.g., Chen X. et al., 2024; Chen Z. et
al., 2024).

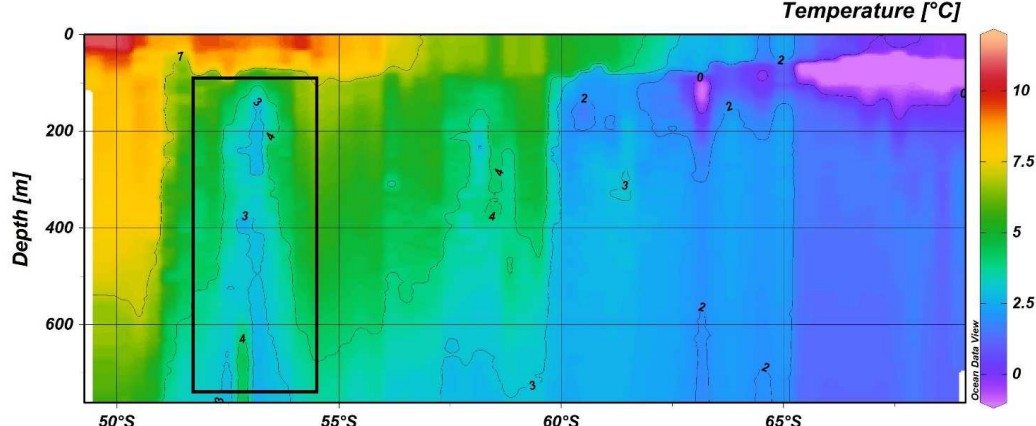


**Figure 6.** Temperature vertical section from XBT data collected during the XXIX PNRA expedition along the New

Zealand–Antarctica "chokepoint" in which the black box identifies the position of an ACC's cold core eddy.

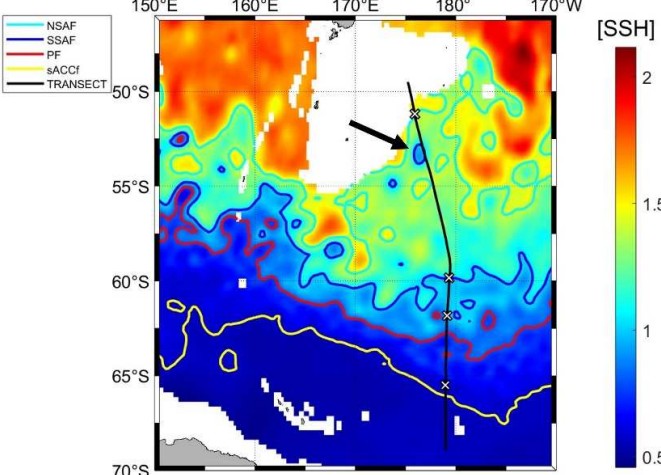


**Figure 7.** Altimetric map of SSH mediated throughout the XXIX PNRA expedition along the PX36 monitoring line.

Contours of different colours identify the position of the main fronts of the ACC retrieved through SSH: NSAF in cyan;
SSAF in blue; PF in red and sACCf in yellow. White crosses represent the position of the fronts derived from XBT data.
The ship's route is represented by the black line. The black arrow indicates the observed cold core eddy.

## 4. Data availability

XBT data are publicly accessible in text format files through the NOAA-NCEI unrestricted
repositories, as listed in Table 6. NCEI (National Centers for Environmental Information) serves as
the official archive for data, metadata, and products collected and provided by NOAA (National





Oceanic and Atmospheric Administration). NCEI also hosts quality checked data by non-NOAA
scientists and is regarded as one of the world's most comprehensive ocean, atmospheric and
geophysical archives, with over 60 petabytes of data covering the full range of Earth's environmental
systems and cycles.
Each XBT file includes the main variables summarized in Table 7, the relative metadata (i.e., probe
type, software, manufacturer, fall rate coefficients, data originator, scientific project, platform) and a
short description of the dataset. One file is created for each research cruise. The naming convention
is xbt_ship_cruise_all_QC, where ship and cruise are identification names of the research vessel used
for the XBT launch and the PNRA research expedition number, respectively, as in Table 1.

**Table 6.** XBT data repository list

| Data set | DOI | Reference |
|---|---|---|
| PNRA X – 1st leg | https://doi.org/10.7289/v5rf5s9v | Cotroneo et al., 2018a |
| PNRA X – 2nd leg | https://doi.org/10.7289/v53r0r5z | Cotroneo et al., 2018b |
| PNRA XI | https://doi.org/10.7289/v5x065b9 | Cotroneo et al., 2018c |
| PNRA XII | https://doi.org/10.7289/v5kd1w6b | Cotroneo et al., 2018d |
| PNRA XIII | https://doi.org/10.7289/v50863mf | Cotroneo et al., 2018e |
| PNRA XIV | https://doi.org/10.7289/v5mg7mtc | Cotroneo et al., 2018f |
| PNRA XV | https://doi.org/10.7289/v56d5r8p | Cotroneo et al., 2018g |
| PNRA XVI | https://doi.org/10.7289/v5s75dpg | Cotroneo et al., 2018h |
| PNRA XVII | https://doi.org/10.7289/v5ng4nzr | Cotroneo et al., 2018i |
| PNRA XVIII | https://doi.org/10.7289/v5qz289c | Cotroneo et al., 2018j |
| PNRA XIX | https://doi.org/10.7289/v5vq3113 | Cotroneo et al., 2018k |
| PNRA XX | https://doi.org/10.7289/v5vh5m45 | Cotroneo et al., 2018l |
| PNRA XXI | https://dx.doi.org/10.25921/hzcp-d813 | Cotroneo et al., 2019 |
| PNRA XXII | https://doi.org/10.25921/c8bm-xh74 | Cotroneo et al., 2018m |
| PNRA XXIII | https://doi.org/10.25921/q29v-c980 | Cotroneo et al., 2018n |
| PNRA XXV | https://doi.org/10.7289/v50r9mmm | Cotroneo et al., 2017° |
| PNRA XXVII | https://doi.org/10.7289/v54j0cbw | Cotroneo et al., 2017b |
| PNRA XXVIII | https://doi.org/10.25921/9YTS-P771 | Cotroneo et al., 2018o |
| PNRA XXIX | https://doi.org/10.25921/220j-b370 | Cotroneo et al., 2024a |
| PNRA XXX | https://doi.org/10.25921/9ph6-c102 | Cotroneo et al., 2024b |
| PNRA XXXI | https://doi.org/10.25921/zf04-ch06 | Cotroneo et al., 2024c |
| PNRA XXXII | https://doi.org/10.25921/vvmp-rr55 | Cotroneo et al., 2024d |
| PNRA XXXIV | https://doi.org/10.25921/jeee-zf77 | Cotroneo et al., 2024e |
| PNRA XXXV | https://doi.org/10.25921/1ysg-dw94 | Cotroneo et al., 2024f |
| PNRA XXXVI | https://doi.org/10.25921/aeg5-hw87 | Cotroneo et al., 2024g |
| PNRA XXXVII | https://doi.org/10.25921/3mmd-tj60 | Cotroneo et al., 2024h |
| PNRA XXXVIII | https://doi.org/10.25921/kte7-d058 | Cotroneo et al., 2024i |
| PNRA XXXIX | https://doi.org/10.25921/jc13-ek97 | Cotroneo et al., 2024l |



**Table 7.** Name and description of the main variables included in the XBT text files.

| Name of variable | Unit | Description |
|---|---|---|
| Cruise | | Cruise name |
| Station | | Identifier number of XBT deployment |
| Time | dd/mm/yyyy hh:mm | Date and time of XBT deployment |
| Latitude [degrees_north] | Decimal degree | Latitude of XBT deployment |
| Longitude [degrees_east] | Decimal degree | Longitude of XBT deployment |
| Temperature [°C] | Celsius degree | XBT temperature measurements |
| Depth [m] | Meters | Depth of the XBT temperature measurements |
| Bot. Depth [m] | Meters | Maximum depth of XBT measurements |
| QF | 0 - 4 | Quality flags of XBT measurements |
| Temperature Corr. [°C] | Celsius degree | Corrected XBT temperature measurements |
| Depth Corr. [m] | Meters | Corrected depth of the XBT temperature measurements |

## 5. Conclusions

The SO is a key place for atmosphere–ocean physical and biogeochemical interactions at different spatial and temporal scales (Falco and Zambianchi, 2011; Cerrone et al., 2017a, b; Buongiorno Nardelli et al., 2017). However, despite their importance, processes in many areas of the SO are still poorly known due to the scarcity of in situ measurements. This is particularly true for the ACC region and its fronts, which are characterized by complex dynamics and intense eddy activity (Trani et al., 2011; Cotroneo et al., 2013; Frenger et al., 2015, Menna et al., 2020; Ferola et al., 2023). To fill this gap, all available measurements provide a significant contribution and should be shared within the oceanographic community.

To this goal, here we present 36 vertical sections of XBT ocean temperature data collected between New Zealand and the Ross Sea (PX36 line) during the Austral summers from 1994/1995 to 2022/2023. This dataset provides direct insights into the 0-800 m thermal characteristics of the Pacific sector of the SO and complements data sourced from observing networks, drifters, ARGO floats and glider fleets. It is also suitable to be combined with enhanced spatial and temporal scale remotely sensed observations and numerical simulations. This comprehensive dataset lays a robust foundation for a nuanced analysis of the key mechanisms governing thermohaline circulation in the SO and for improving our knowledge of the physical and biogeochemical characteristics of the four-dimensional ocean.

The continuation of this XBT collection over time, in the framework of the Italian PNRA research expeditions to Antarctica, is particularly important due to the inherent challenges associated with data



acquisition in the SO and promises an increasingly comprehensive and detailed understanding of thermal variations in this specific maritime region.

**Author contributions.** GA, YC and AIF conceived and designed the manuscript. GA, YC, PC, PF, GF, GB, NK, GS, EZ and AIF collected the measurements and organized the XBT dataset. GA, YC, LF and AIF carried out the quality control analyses. All authors analysed the achieved results, contributed to the writing, and approved the final manuscript.

**Competing interests.** The authors declare that they have no conflict of interest.

**Acknowledgements.** This study was made possible thanks to the contribution of the Climatic Long-term Interaction for the Mass balance in Antarctica (CLIMA), Southern Ocean Chokepoints Italian Contribution (SOChIC), Marine Observatory of the Ross Sea (MORSea), Effects of the east current on the Salinity variability in the Ross Sea (ESTRO) and Physical and biogeochemical tracing of water masses at source areas and export gates in the Ross Sea and impact on the Southern Ocean (SIGNATURE) projects, part of the Italian National Antarctic Research Program (PNRA). Special thanks go to Arturo De Alteris, Massimo De Stefano and Giovanni Zambardino who provided essential support to data acquisition, as well as to the captain, officers, and crew of the research vessels used for XBT launches.

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
