# Peer review of "XBT data collected along the Southern Ocean "chokepoint" between New Zealand and Antarctica, 1994-2024"

_Earth System Science Data, 2024_

## Referee Comment (RC2)

[referee-annotated manuscript omitted]

---

## Author Response (AR1)

Dear Editor,

We sincerely appreciate the valuable comments and suggestions provided by the referees, which have been extremely helpful in improving our manuscript. Below, we present a detailed point-by-point response to all referee comments and outline the main revisions made in the manuscript. These changes are also tracked in the marked-up version. For clarity, referee comments are presented in blue, authors' response in black.

We hope our revisions adequately address the referees' concerns and look forward to your feedback.

Best regards,

Giuseppe Aulicino and co-authors

**Reviewer 1**

Summary: The paper nicely describes a valuable dataset of XBT observations across the Southern Ocean from New Zealand to the Ross Sea, Antarctica. These data cover 1994 to 2024 along the PX36 Ship of Opportunity (SOOP) line. The quality of the data is described and suitable QC procedures followed to improve the dataset for end users. Depth corrections have been applied appropriately. The authors have done an excellent job in summarising the data collected and presenting examples of the use with eddy and front identification using complimentary datasets such as satellite altimetry and Argo datasets. I recommend publication after some corrections to improve the manuscript (see below) and access to the data could be improved given the current situation post-hurricane in the USA where the NCEI servers are difficult to access. Similarly, publication of some code to access and display the data would be useful to go along with the manuscript.

We are thankful for the received comments and accepted all the proposed suggestions to improve the manuscript as detailed below.

Introduction: I recommend including reference to the GOOS Ship of Opportunity program (SOOP) and the SOOPIP (Ship of Opportunity Implementation Panel) in the manuscript. Certainly the authors are part of these programs and the data makes a significant contribution to the SOOP program.

Reference to the GOOS SOOP and SOOPIP have been added in the Introduction (see lines 80-86):

"The Global Ocean Observing System (GOOS) Ship Of Opportunity Program (SOOP), and the related Ship of Opportunity Program Implementation Panel (SOOPIP), address scientific and operational (standardization, maintenance, and advancement of the instruments and techniques) goals, respectively, to building a sustained ocean observing system, e.g., supplementing dedicated research vessels in the collection of upper ocean in situ XBT data through the use of ships that are already traversing the world's oceans (Legler et al., 2015; Goni et al., 2019)".

Line 109, suggest the reference should be Cowley et al 2013 as it refers to depth and temperature variations over time.

Done.

Table 2 contains Hanawa 1995 FRE values for the T7 probe, not manufacturer FRE as stated in the title. If the Hanawa FRE is used, please include in reference. Also confusion over which FRE the non-corrected Depth values use in the dataset (see comments in the data section below).

Many thanks for your suggestion. The revised Table 2 contains only the physical characteristics of the XBT probes used in this study (Sippican T5 and T7). The FRE coefficients provided by the manufacturer were removed from the table to prevent potential confusion for the readers. A full description of the FRE used in this study (i.e., Hanawa et al., 1995) is now discussed in Section 2.3 (XBT data biases correction).

Line 155. Suggest including Cowley & Krummel with Parks et al reference.

Done.

Line 187, 202, 206: Suggest 'log sheets' or similar in place of 'launch clipboards'.

Done.

Line 208. Grammar: 'resulted the most correspondent' perhaps should be ' corresponded'

Done.

Line 230. Grammar: 'unusally' should be 'unusual'

Done.

Line 276-277. The reference for nodc has an out-of-date table. I think the authors have used the most recent update for the Cheng correction, but these tables do not contain that update. Perhaps the authors can ask Cheng to provide a link to the latest tables. Also, the second link to iap data is to the global ocean dataset, not the XBT correction tables.

Many thanks for your suggestion. The manuscript has been improved with the correct (direct) links as follows (see lines 288-292): "A full description of the methodology is available at https://www.ncei.noaa.gov/products/xbt-corrections (see CH Correction Method); the update tables of applied coefficients are available at http://www.ocean.iap.ac.cn/ftp/images\_files/CH14\_description/CH14\_table1\_update2023.txt and http://www.ocean.iap.ac.cn/ftp/images\_files/CH14\_description/CH14\_table2\_update2023.txt".

**Line 302. What is 'SBdy'?**

Modified to "Southern boundary of the ACC"

Line 319. Grammar: please check the sentence as it could be improved to make clearer the author's meaning.

Sentence modified as follows: "This is highly desirable in regions significantly influenced by topographic steering, such as the area south of New Zealand, where the presence of the Campbell Plateau strongly affects the ACC path (Figure 5)."

Section 2. I think you need a subsection here describing how you created the section plots. What algorithms did you use to make these? Are the section data available for the user to download? Can you include the code in a python notebook or similar so the user can create them?

Thank you for your comment. The format and labels of the presented XBT dataset are specifically designed to ensure compliance with Ocean Data View (ODV), enabling seamless upload and ease of use. The interpolated section data are not available for direct download. However, to enhance reproducibility, additional details about the parameters used to create the ODV interpolated temperature section have been included in the manuscript at lines 303-307:

"All the temperature sections presented in Figures 4, 6, and S1-S36, were realized using ODV software and applying consistent interpolation parameters. The adopted zonal interpolation is based on a spatial weighting model that incorporates three temperature profiles (a central reference profile, an upstream profile, and a downstream profile), considering a maximum influence range of 60 km along the zonal direction and 20 m along depth".

Moreover, based on the Reviewer 1 helpful suggestion, we have also added a Python script for basic data visualization (i.e., latitudinal temperature section and vertical temperature profiles). This information is now reported in section 4 (lines 410-412) as follows:

"Additionally, a Python code for basic XBT data visualization is included in supplemental material S37 (such as shown for scatter plots of vertical temperature profiles and latitudinal temperature sections in Figures S38 and S39)."

**Data comments:**

I am unable to download the data via https, but was able to access files via ftp (although this is a bit slow). There is some difficulty accessing the servers at NCEI. Can you provide an alternative route to the data? Perhaps a Zenodo link or another DOI to the full dataset (one link) would be useful in addition to the individual NCEI links.

Data are now fully restored at NCEI repository. Additional efforts to ensure continuous availability are under way at NCEI after their servers have been severely impacted by Hurricane Helene in September 2024. Moreover, an improved version-3 of the XBT dataset will be soon available, including corrected products, detailed information about the applied FRE coefficients and additional metadata (following EMODnet guidelines). Nevertheless, as suggested by Editor and Reviewer 1, the full XBT dataset presented here is now publicly accessible also through an alternative route on the Zenodo archive, at https://doi.org/10.5281/zenodo.14848849

This information is reported in section 4 (lines 390-400) as follows:

"The full XBT dataset presented here is publicly accessible as text format files at https://zenodo.org/records/14848849. Individual cruise data files are also available through the National Oceanic and Atmospheric Administration (NOAA) National Centers for Environmental Information (NCEI) unrestricted repositories, as listed in Table 6. NCEI serves as the official archive for data, metadata, and products collected and provided by NOAA scientists. Additionally, NCEI hosts quality checked data from non-NOAA scientists, which must go through a scientific appraisal process before being accepted into the archive. For this reason, our XBT data underwent a thorough review and improvement process (see sections 2.2 and 2.3) prior to publication, resulting in the version-3 products. Nevertheless, as noted above, the full dataset presented here is also available through the Zenodo repository, providing an alternative access point in case of difficulties retrieving the single-cruise information from NCEI".

**I reviewed the xbt\_araon\_XXXIV\_all\_qc.txt file.**

1. Fall Rate Coefficients listed in the file (manufacturer T7 coefficients) do not match the ones in the paper (Hanawa 1995 coefficients). Please check which coefficients were used as this will make a difference to the Cheng 2014 corrections applied.

The Fall Rate Equation coefficients used for temperature and depth bias correction (i.e., Hanawa et al., 1995 coefficients, as suggested by Cheng et al., 2014) are now reported in each XBT file. However, the coefficients provided by the manufacturer are also provided in the metadata, allowing anyone who wishes to recalculate the corrections in a different way than using Cheng et al. (2014).

Actually, all the XBT files include:

a- the time elapsed since the probe's release;

b- the depth derived from the elapsed time using the Manufacturer Fall Rate Equation Coefficients;

c- the depth derived from the elapsed time using the Hanawa et al. (1995) Fall Rate Equation Coefficients;

d- the depth corrected applying Cheng et al. (2014) to "c", using Hanawa et al. (1995) Fall Rate Equation Coefficients;

e- the temperature measured by the probe;

f- the temperature corrected following Cheng et al. (2014) with Hanawa et al. (1995) coefficients

Table 7 was updated accordingly, as well as data description in each XBT cruise dataset. Additionally, XBT metadata were improved to meet EMODNet requirements as reported in section 4 (see lines 401-404):

"Each XBT file includes the main variables summarized in Table 7, the relative EMODNet-compliant metadata (i.e., about probe type, software, manufacturer, data originator, scientific project, platform, uncertainties, QF code), detailed information about the FRE coefficients used for temperature and depth bias correction described in section 2.3, and a short description of the dataset."

2. Label 'Bot. Depth [m]' is not the bottom depth, but the XBT maximum depth reached.

Label 'Bot. Depth [m]' (ODV compliant label) is now described in the XBT file as "Maximum reached depth of the XBT probe".

3. I loaded up the data and plotted it using Python and found the quality flags are well applied, however there are some bad data remaining (hit bottoms and some leakage) in a few of the profiles. Generally, the QC looks acceptable for the one file I reviewed.

Additional visual check was provided for identifying remaining bad data.

4. It would be very helpful and enhance the paper if you could provide some code to read and plot the data for the users. A python notebook would be very helpful. Below is the basic code I used to load and look at the data. You could expand on this.

We thanks the Reviewer for the code. We expanded on that one to provide a Python code for a basic visualization of the data, including a latitudinal temperature section and a scatter plot showing the vertical temperature profiles.
* * *
**Reviewer 2**

Very useful paper presenting XBT data collected in under sampled region of the South Ocean over 30 years of fields expeditions. Please see my remarks to the main text of the manuscript in the supplemented file.

We are really thankful Reviewer 2 for the comments. We accepted all the suggestions provided in the supplemented file to improve the manuscript. Changes are described in the point by point reply below.

Line 127: "sampling rate" changed to "sampling interval"

**Line 138: Explain ZAMAK**

Zamak is a family of alloys primarily composed of zinc along with aluminium, magnesium, and copper. The name "Zamak" is an acronym derived from the German words for its constituent metals: Zink (zinc), Aluminium (aluminium), Magnesium, and Kupfer (copper). This information is included in the caption as follows

"Table 2. Characteristics of the different XBT probes used in this study: nominal depth guaranteed by Sippican; maximum ship speed suggested by Sippican for an optimal drop; amount of ZAMAK (a zinc-based alloy enriched with aluminium, magnesium, and copper), copper and plastic for each probe type (adapted from Simoncelli et al., 2024)".

Line 205: As suggested, "elevations" is changed to "*depth*".

Table 4: As suggested, we use the term "range" in place of "threshold". We went through the text for re-wording where it was needed. The ranges were defined through the use of ARGO data collected in the study area between 2004 and 2023 as reported in the text (lines 214-215):

"The ranges were defined through the use of ARGO data collected in the study area between 2004 and 2023".

Line 302: "SBdy" changed to "Southern boundary of the ACC".

Figure 4: Please provide interpolation settings and add data points on the plot. Section map insert (or additional plot) would also be helpful

As suggested, data points have been added to the plot, as well as the section map. Please note that all the XBT sections have been improved using new interpolation parameters as detailed below. Information about interpolation settings is provided at lines 303-307:

"All the temperature sections presented in Figures 4, 6, and S1-S36, were realized using ODV software and applying consistent interpolation parameters. The adopted zonal interpolation is based on a spatial weighting model that incorporates three temperature profiles (a central reference profile,

an upstream profile, and a downstream profile), considering a maximum influence range of 60 km along the zonal direction and 20 m along depth.".

**Lines 318-321: Please rephrase**

**Sentence rephrased as**

"This is highly desirable in regions significantly influenced by topographic steering, such as the area south of New Zealand, where the presence of the Campbell Plateau strongly affects the ACC path (Figure 5)."

Lines 339-341: Sentence improved as suggested:

"The eddy is characterized by a maximum negative temperature anomaly (eddy's core) of about - 4°C compared to the surrounding water."

Figures 5 and 7: Cyan line does not looks like front in this case. If it is an indicator of the front position, maybe it should be generalized

Fronts contour lines have been improved.

My concerns are about section plots presented. I recommend to review the interpolation parameters used for section plots presented in the 'Supporting Information' document, in order to avoid some artefacts, which I believe are caused by too small search radius set during section creation. It is also important to use same interpolation setting for all section and provide this information (i.e. horizontal search radius and vertical search radius used for interpolation in ODV) in the text. It also will be helpful to add a profiles markers on the section plots as well as section boundaries set in ODV for each section (this will allow to access spatial limits during interpolation). It will be useful to have plots in the 'Supporting Information' document numbered individually (like S1-1, ..., S1-35), - it will make reference to some individual plot more convenient. As for now all 36 section plots are referred to as Fig S1, therefore, below, I use page numbers to refer to some particular plots.

Remarks to the section plots in the 'Supporting Information' document:

I believe some of the features on the plots appears as result of using too short horizontal search radius, so interpolation algorithm is only using data from one profile not taking into account its neighbors. Provided that distance between profiles (n) is about ~20 km, I would recommend to employ 3n value, i.e. ~60km. I only mention few pages/section plots below, but these remarks are related to all section plots - there are many gaps in the stations position which could be the cause for the interpolation artefacts now existing everywhere.

As suggested, all the section plots have been reviewed using the same interpolation parameters in ODV. The adopted zonal interpolation is now based on a spatial weighting model that utilizes three temperature profiles (3-n values): a central reference profile, an upstream profile, and a downstream

profile considering a maximum influence range of 60 km along the zonal direction and 20 m along the depth. This information is also provided in the 'Supporting Information' document.

Additionally, profile markers and section boundaries have been added to the plots.

As requested, plots in the 'Supporting Information' document are now numbered individually (S1-S36).

---

## Author Response (AR2)

Dear Editor,

We accepted all the minor suggestions provided by Reviewer 2 and improved the manuscript accordingly at lines 113-117 and 280.

Best regards,

Giuseppe Aulicino and co-authors